# Adaptive sport as affirmation: "We focus on our strengths, not our disabilities"

**Norah Anita Schwartz**⬤*, **Christine Alysse von Glascoe**

Departamento de Estudios Poblacion, Secion de Salud Publica, El Colegio de la Frontera Norte, Tijuana, Baja California, Mexico

* Schwartz@colef.mx

## Abstract

This paper offers a counter-narrative to the stereotype of people with physical and cognitive impairments being less inclined to participate in athletic activities. It contributes to the affirmative model proposed by Swain and French, which posits a non-tragic view of disability that encompasses positive social identities. We employed the tools of ethnography and phenomenology to explore the adaptive athlete experience among individuals practicing various sports and exhibiting divergent levels of proficiency. Findings confirm the appropriateness of the affirmative model and provide examples of movement from the state of liminality to that of *communitas* as defined by Victor Turner.

## Introduction

Breaking with societal boundaries and norms, the adaptive sports community has been attaining goals not traditionally expected of people with physical or developmental challenges. In this article, we explore the culture of adaptive athleticism as fitting the affirmation model of disability proposed by Swain and French [1] and the movement from liminality to *communitas* as defined by Victor Turner [2]. The affirmative model, which modifies the limiting focus on disability and impairment to include possibilities, was a breakthrough in models of disability when it was proposed by Swain and French in 2000 [1]. Ethnographic and autoethnographic research in the domains of adaptive athletics and community confirms the viability of the non-tragic view of impairment set forth in the affirmation model.

Our research also contributes to the theory of community of practice [3] as it applies to adaptive athletes. As one moves from the sense of being alone and excluded to that of being accepted by a community, they enter the new phase of *communitas*. Our results show how the adaptive sports community fits the definition of community of practice set forth by Lawthom, Lave and Wenger, as "a collection of people bound together by purpose, activity, values, desires or, perhaps, labels" [4]. In the broadest sense, the community of adaptive athletes consists of persons of any age with any type(s) of impairment who engage in an adaptive sport, as well as those who sponsor and support their engagement.

As part of a larger ethnographic project, the research questions aim to address the role of athletics in the lives of adaptive athletes. Specifically, we aim to understand the unique challenges, benefits and personal growth experiences that participants face in their chosen sport(s).

**Data Availability Statement:** A minimal data set underlying the results described in our manuscript has been deposited and under review at Qualitative Data Repository: 59Schwartz, Norah2023-02-102023DRAFT VERSIONDisability SportQDR Main

Collection Https://doi.org/10.5064/F65BT3T9doi/
10.5064/F65BT3T9.

**Funding:** The author(s) received no specific
funding for this work.

**Competing interests:** The authors have declared
that no competing interests exist.

What challenges, for example, does a newly disabled person face when entering or re-entering athletics? How does participating in team sport alter one's self-perception and view of life? For parents of children with impairments, how does being a part of an adaptive athletic community help their children grow and thrive? What is the range of emotions experienced by athletes and their families?

## Background

Social stigma, pity, marginalization, 'othering', and infantilization have been imposed on the body-minds of individuals with disabilities throughout history [5–7]. Today, society is being challenged to reconfigure its relationship with impairment, disability and ableism [8]. Progress toward athletic equality for people with disabilities, the subject of this paper, has been slow but steady [9, 10].

In 1978 the UNESCO International Charter of Physical Education and Sport declared that access to physical education and sport is a right that should be guaranteed for all human beings. The United Nations Convention on the Rights of Persons with Disabilities (UNCRPD) [11] urges States Parties to "take appropriate measures" to enable "persons with disabilities to participate on an equal basis with others in recreational, leisure and sporting activities", and to "encourage and promote the participation, to the fullest extent possible, of persons with disabilities in mainstream sporting activities at all levels". As recently as 2015, the UNESCO charter was revised to specifically address gender equality, non-discrimination, social inclusion, and benefits to persons with disabilities through sport.

Herein, we distinguish the term 'dis/ability' from the narrow dictionary definition of disability as "a physical, mental, cognitive, or developmental condition that impairs, interferes with, or limits a person's ability to engage in certain tasks or actions or participate in typical daily activities and interactions" [12]. We use the term 'impairment' to refer to a physiological condition, as contrasted with the interchangeable use of 'disability' and 'dis/ability' to connote the societal imposition on people with impairments that limit, exclude, or oppress one's ability to fully participate in society.

Anthropologists with impairments are often left out of the theoretical inquiry on dis/ability, even within the field of anthropology. Erin Durbin [13] emphasized the importance of aligning anthropological rigor in the 'unsticking' of this unconscious bias:

Our "disabled"—crip, blind, mad, slow, neurodivergent, creaky, deaf, weird, obsessive, freaky, exhausted, chronically ill, leaky, depressed, interdependent—bodyminds invigorate anthropological inquiry and theory unknowingly stuck in well-worn ableist grooves.

Anti-ableism requires a repositioning away from the objectification of individuals living with disability towards being able to 'step into their world'. Since the early days of the disabilities rights movement, advocates and activists have insisted on gaining a seat at the table in order to (re)present our own bodyminds. This paper presents voices of a new generation, one which insists not only on having a place at the table, but also on the podium.

Arguing the point of thinking simultaneously about disability *and* ability, Dan Goodley [14] questions the meaning and value of being 'able'. He "wants to think through some tricky ideas. . . What do you mean by being able? What is valued by being as possibly or ideally able as one could be?" By splitting the term 'dis/abled', Goodley recognizes that people find it difficult to define 'normal' and 'ability', yet are far more ready to categorize 'abnormal' and 'disability'. The concepts of dis/ability and impairment, and the lives of individuals and families dealing with these impairments, encompass so much more than that, as evidenced by the

foundational ethnographic writings of Joan Ablon [5, 15, 16], Devva Kasnitz and Russell Shuttleworth [17], and Faye Ginsburg and Reyna Rapp [18], as well as the applied bioengineering research of Hugh Herr [19, 20].

David Purdue and P. David Howe [21] maintain that *all* bodies possess limitations and need to be seen in the context of the sport in which they compete. Many people still find it surprising to see someone with any kind of evident impairment competing at high levels in sport. In their paper, 'The In/validity of Supercrip' [22], Carla Filomena Silva and P. David Howe suggest that the iconography of the supercrip in sports continues to exist because the expectations of achievement are so low. The late comedian and disability rights activist Stella Young [23], coined the term "inspiration porn" to define an expectation by non-disabled people to be inspired by those with disabilities simply for "their ability to get out of bed in the morning and remember their own name". Conversely, Silva and Howe [22] warn that 'supercrip' narratives—those of the glorified and inspirational disabled person—may "have a negative impact on the physical and social development of disabled individuals by reinforcing what could be termed 'achievement syndrome' or 'success *in spite* of disability'. Describing the "supercrip" as the flip side of the pitiable poster child, Joseph Shapiro [24] writes that:

> the inspirational disabled person. . .is another model deeply moving to most nondisabled Americans and widely regarded as oppressive by most disabled ones. The disability rights movement discards the notion that people with disabilities should be courageous or heroic superachievers, since most disabled people are trying simply to lead normal lives, not inspire anyone.

Often held to higher standards as 'role models' and fodder for the media, adaptive athletes are subjected to the same stereotypes and social challenges as non-athletes with physical or mental impairments. On this issue, poet Eli Clare [25] shares:

> Supercrip stories never focus on the conditions that make it so difficult for people with Down's to have romantic partners, for blind people to have adventures, for disabled kids to play sports. I don't mean medical conditions. . .I mean stereotypes and attitudes. I mean oppression.

## Adaptive athletes as counter-narrative

Within the adaptive athletics world, however, there *does* remain both a need and a place for inspiration, role models and mentorship. Despite, or maybe because of, overwhelming physical and societal barriers, challenged athletes continue to break through what has been thought of as the barriers of human possibility. Blind alpinist and adventurer Eric Weihenmayer, arguably one of the most accomplished athletes in the world today, was the first blind person to climb Mt. Everest. He later completed the Seven Summits, solo-kayaked down the Colorado River, and was one of few competitors to complete the Primal Quest Challenge. Weihenmayer dubbed those who would comment that "even *I* wouldn't do that stuff", as having the "Even I Syndrome" [26]. Together with Mark Wellman and Hugh Herr, he founded the No Barriers leadership program.

Says Sarah Reinersten, the first above-the-knee amputee to complete Hawaii Ironman, "The difference for me came when I met other amputees. I think that sometimes we need to see ourselves in that reflection of possibility" [27]. Other notable firsts include: Jim MacLaren, quadriplegic; Hawaii Ironman; world record holder; founder of the Challenged Athletes Foundation [28, 29]); Hugh Herr, prodigal rock climber; bionics inventor [30]; Rudy Garcia-

Tolson, bilateral amputee; most decorated Paralympian triathlete [31, 32]; Rick Hoyt, spastic cerebral palsy, completed 1000+ marathons and triathlons; educator [33]; Steve Wampler, first person with cerebral palsy to summit El Capitan; founder of Camp Wamp [34], Chris Nikic, first person with Down Syndrome to complete Hawaii Ironman; author, public speaker and entrepreneur [35] and Mark Wellman, the first wheelchair athlete to summit both Half Dome and El Capitan; co-founder with Erik Weihenmayer and Hugh Herr of No Barriers leadership program [36].

A study on motivation in adaptive sports found that, beyond its physical benefit, participants experienced transformations by transcending a disability identity through the perception of self-efficacy that allowed some to establish or maintain a self-concept of being competitive, risk-taking, or athletic [37]. Another study found that adaptive sports involvement provided participants with opportunities to build social networks, experience a sense of freedom and success, positively compare themselves with others without disabilities, and feel a sense of normalcy [38].

Goodley [39] insists that: "any study of disability has to begin with the accounts and aspirations of disabled people. This, for me, is a non-negotiable starting point." Disability activist and scholar, Sara Goering [40], reminds us that, "for many people with disabilities, the main disadvantage they experience does not stem directly from their bodies, but rather from their unwelcome reception in the world, in terms of how physical structures, institutional norms, and social attitudes exclude and/or denigrate them." A growing component of athletes construct a counter-narrative defying the misconception that those with disabilities tend not to participate in physical activity [1, 41]. The affirmative model constructively encompasses individual and collective positive social identities grounded in the benefits of lifestyle along with the life experience of being impaired and disabled. Says Weihenmayer, "I'm not just doing these things so I can prove that blind people can do this or that. That's kind of shallow. . .You do it because that's living fully" [42]. Yet, a world class athlete may climb the highest mountains and still be stared at in the grocery store; not for achieving great accomplishments, but for using a cane.

## Liminality and *communitas*

The concepts of liminality and *communitas*, or the stages of 'betwixt and between' that comprise the pre-liminal, liminal, and post-liminal (reintegration) states of being, as chronicled by Turner [2], provide an apt tool for describing the process of moving from an interstitial state of isolation to that of discovery and acceptance as part of an adaptive athlete community. Sudden or progressive changes to one's body can result in an embodied sense of isolation and dysmorphia. This sense of loneliness can be overpowering for individuals and families dealing with a chronic or degenerative illness, as well as for those adjusting to recent impairment [43–45].

Turner [2] describes liminality as representing a state of flux and can be seen as a rite of passage that often results in an existential state of *communitas*, or the sharing of common experience. For challenged/adaptive athletes, the rite of passage can take place in many forms, including that of participating in one's first athletic competition—the underlying theme being a desire to be part of a community that has more often than not been relegated to the respective margins of societies. In her dissertation on liminality, Molly Bloom [46], for example, a wheelchair basketball player herself, demonstrated how wheelchair basketball players construct narratives of competence, in contrast with common stereotypes that emphasize "overcoming", and even "passing", as able-bodied. She maintains that in so doing, the players construct a shared sense of identity, making such "narratives of competence" a

more productive theoretical framework for understanding the stories of disabled athletes than 'overcoming narratives', because the storytellers are not attempting to pass as able-bodied, but rather, they are presenting themselves as capable people, who are not trapped in liminal space due to their disabilities.

The deep understanding and experience of the liminal state can become the impetus for the bond that is developed through mutual support and competition in what Reinertsen [27] describes as "push[ing] for those crazy finish lines". Born with proximal femoral focal deficiency, which stunted the growth of her left leg, Reinertsen wanted desperately to fit in with her schoolmates but had been denied opportunities to participate in sports as a child. Having decided in favor of "liv(ing) to make a difference", at the age of 30 she became the first above-the-knee female amputee to complete the onerous Hawaii Ironman. In her own words: "There's still that little seven-year-old girl inside of me that's trying to prove to the world that a person with a disability deserves to be on the soccer field or at the marathon starting line." On mentoring a young girl, Reinertsen states:

> I want to be her role model, to show her that she doesn't have to be on the sidelines. She can climb mountains, she can ride bikes, she can swim in the pool, she can go the distance. That's part of why I do this—to show. . .what's possible—to push for those crazy finish lines. But it's only crazy till you do it. [27]

### Liminal bodies/nanotechnologies

Sociologist Irving Kenneth Zola proposes that. . . "we are entering an era where almost every human body part and function becomes replaceable or, at least, 'assistable' by some technical device" [24]. It has been nearly 30 years since Shapiro [24] noted that "it is not uncommon for paraplegics to believe they will walk again, despite all evidence to the contrary. It is a way to hide from the stigma of disability." He credits adaptive athletes, including Jim MacClaren [28, 29]), founder of the Challenged Athletes Foundation, for breaking through stigma and furthering the marvel of technological advances that are in use today by athletes without requiring them to 'walk again'.

Advancing technology is contributing to greater accessibility for athletes to participate in activities previously reserved for non-disabled bodies. According to a review article by Tan [47]: "Prosthetic limbs made of nanomaterials can significantly improve the quality of life of the disabled due to their unique. . . capabilities". Vibrotactile feedback systems are creating a space for adaptive athletes with prosthetics to perform as well or better than non-disabled athletes [48]. Hugh Herr, the father of bionics, who lost both his legs in a hiking accident at the age of 17, expects to see an end of disability in his lifetime. However, these advancements bring forth many ethical issues. Because such "novel neurotechnologies may alter human identity and society in profound ways", Goering and Yuste advocate for the early integration of ethics into this promising field [49].

The advents of nanotechnology and bionic prosthetics also contribute to the textured liminality of bionic bodies outperforming non-bionic bodies, adding to the controversy over integration, or lack thereof, of adaptive athletes in the realms of competitive sport. Herr has devoted his life to advancing "design technology that normalizes or extends human physicality." Says Herr:

> As a society, we can achieve these human rights if we accept the proposition that humans are not disabled; a person can never be broken. Our built environment, our technologies are broken and disabled. We the people need not accept our limitations, but can transcend

disability through technological innovation. Indeed, through fundamental advances in bionics in this century, we will set the technological foundation for an enhanced human experience, and we will end disability. https://www.youtube.com/watch?v=CDsNZJTWw0w [30]

The question remains whether athletes with technologically advanced prosthetics will, despite the urging of the UNCRPD to ensure equal access to mainstream sporting activities at all levels [11], be disqualified from future events due to having a perceived 'unfair advantage' over those without the need or use of such devices.

## Methods

### Participants

The purposive sample was complemented by a snowball selection consisting of male and female adaptive athletes who were chosen to represent a wide variety of impairments; including congenital malformation, sudden disabling event, or chronic/degenerative illness. Activities were chosen based on those sponsored by local adaptive athletic organizations; including, rock climbing, surfing, and triathlon. Determination of dis/ability included self-reports of a physician's diagnosis or on self-perception of impairment. All participants were asked to discuss the role of athletics in their lives and the unique challenges and benefits of participating in their chosen sport(s). Interviews were audio-recorded and transcribed verbatim using Otter.ai.

### Study design

A reflexive phenomenological study approach [50–52] was used in order to garner a deep understanding of the meaning and effects of athletics in the quotidian lives of individuals living with impairment, and to gain a perspective on what Ginsburg and Rapp refer to as "everyday life with a difference" [18]. Autoethnographic field notes and photographs were recorded during various events and training sessions by one of the authors (NS) who participated as an adaptive athlete team member and volunteer. Personal reflections in her voice are included throughout this account. The ability to share similar understandings, especially through listening to stories of impairment and traumatic mobility-altering experiences [53, 54] can promote mutual acceptance which, in turn, can lay a foundation of trust that might otherwise be difficult to develop with project participants [55]. Employing an entangled ethnographic approach, as defined by Ginsburg and Rapp [56], and intersubjective phenomenological analysis enabled the researchers and participants to explore how they perceive, think, know and imagine their impairments, and the effects on their lives and those around them.

There were no risks to participants, other than potential loss of anonymity. Data were anonymized by using pseudonyms and asking only the first names of participants where appropriate. Written or verbal approval was given to record the interviews and the final paper was reviewed by a sample of participants.

### Setting

Unless otherwise noted, the setting takes place in indoor rock climbing gyms, outdoor climbing venues, beaches and individual homes in Southern California, United States.

### Data collection

The purpose of the study was explained to the participant and written informed consent was requested to record the interview and to take photographs. Anonymity was guaranteed.

Nevertheless, upon reading a draft of the article, a number of members of the climbing team requested that their first names and photographs be included. Interviews were conducted in the venue of choice of the participant. A copy of the transcript and any photographs that had been taken were offered to the participant. As this was an unfunded project, no compensation was offered.

## Data analysis

Our analysis draws from interviews, participant observation, and autoethnographic self-reflection enhanced by thematic conversations between the co-authors. These dialogues were enriched by the "insider-outsider" perspectives provided by one of the co-author's dynamic and ongoing engagement and participation as an active member of the adaptive athletic community.

Recorded interviews were transcribed, and their verbatim content discussed between co-authors. Data were organized and coded according to repeated themes and phenomena [57]. Open coding [58–60] was performed to search for patterned responses [61], and to produce textual elements to explain the data. Data were iteratively collected and analyzed between events to enable a "reflexive and interactive" treatment of the data [50]. Field notes and interviews were interpreted phenomenologically by key categories [62]. Approval was granted by the Internal Review Board of the Colegio de la Frontera Norte, Tijuana, Mexico.

## Results

### Cimb on: "I have a disability, but I'm not disabled"

Adaptive athletes focus 'on our strengths, not our disabilities', placing abilities ahead of impairments. Occupational therapy student, Coach Devon, noted that with "adaptive climbers. . .we focus on what climbers can do and where they may have limitations due to abnormalities." In order to progress to a competitive level, consistency in any sport is imperative; thus, the first adaptive climbing team in Southern California was formed to meet three days a week for two to three hours at a time. The team soon began bringing home medals and was featured in an award-winning documentary filmed by intellectually and developmentally disabled film students—though we prefer to simply refer to them as gifted film makers. The film, CLIMB, premiered at the San Diego Museum of Photographic Arts in 2022 and was awarded the Best Film: disABILITY Education Film Festival 2022 and Award of Merit—Disabilities Category: *Accolade Global Film Festival* 2021, among others [63, 64].

Not fitting into either the objectified pitiable pathologized poster child camp [18], nor that of the stereotypical supercrip [24], adaptive athletes prefer to be recognized for their individual and team athletic endeavors and accomplishments. Jillian, a professional ballet dancer turned rock climber and coach emphasized that, "I have a disability, but I'm not disabled":

I've been really upfront about my disability. . . You have to surround yourself with the right people. This whole movement is so important. Be with people who know where you're at. It's hard. Being in an adaptive community is so important.

Jillian conceived the idea of forming an adaptive climbing team while observing a child in a wheelchair being told that she was "not going to do that. . .your sister is". Frustrated, too, that adaptive, or para climbing was only offered infrequently in most climbing gyms, she ensured that her team would have the professional training and competitive opportunities they deserve. Currently, adaptive rock-climbing teams and groups are available throughout the world, and may be included as a sport in the Paralympics for the first time in 2028. The team portrayed in

our study had been looking forward to hosting the US national adaptive rock-climbing competition in March 2020 immediately before the gym shut down due to the Covid-19 pandemic. It has since resumed training for national and global competitions.

James (pseudonym), who had never competed before, was expected to be one of the rising stars of competitive adaptive climbing. Born in Thailand with a congenital birth defect resulting in three incomplete limbs, he chose to have his short leg amputated and replaced with a prosthesis. Upper body prosthetics proved cumbersome and did not aid in his climbing or daily activities, so he preferred to not use them. A fierce climber, James attempted the most difficult routes in the gym, calling much attention to himself. At 21 years old, his ambition was to become a coach and train other adaptive athletes; however, learning disabilities interfered with his finishing high school, thus, hampering his ambitions.

An extrovert by nature, James quickly made friends and climbed with some of the best non-adaptive climbers in the gym. Yet the team was his anchor:

NS: What are your plans for the competition?

I plan to work extra hard every single time before competition starts. Do better every single time. Work out more; but more warm-up before I start to climb. . . I don't know how many routes I do in a day. I just try them all.

James would "try to come 45 minutes early before practice starts. Stay an extra hour. Do more climbing." Sometimes he would talk about Thailand with a longing to return combined with an ambivalence based upon his awareness that access to adaptive athletics would be limited. He perceived that he would be considered "different" there; subjected to what his adoptive mother referred to as "the poor little thing" syndrome:

Since I moved to California, everything changed in my life. I feel more connected being part of a team. . .I want to find my own path. My dream: I want to be a coach in another country—for disabled sports. . .There are so many people in other countries. . .that don't have the same opportunities as here in California. . .I want to break the stigma. . .I can do everything on my own. . . I just want to give something back to my family and community.

For the time being, James has returned to Thailand during the pandemic without realizing his dream of becoming a coach.

Jono, a participant from New Zealand who uses a prosthetic leg, enjoys climbing outdoors with his friends. Like Jillian, Jono does not "really see myself as having an impairment. . .It is not framed like that in my life. . .Disability has so much nuance. . .There is a glorification of people with disabilities" Comparing New Zealand to the United States, he relates that in New Zealand, prosthetics are freely available, though of lesser quality. Referring to disparities in access, Jono relates a surprising statistic:

[worldwide], nine out of ten amputees do not have prosthetics. . .In the US, many people just have access [to low-cost prosthetics] . . . they rely on the generosity of the prosthetic companies. In 25 to 30 years, I would like to see everyone have the same access; though something would have to change (Fig 1).

Tanner, who won second place in his division at the 2021 Paraclimbing World Championships, in Moscow, Russia, and first place in the USA national competition in 2023, is not only optimistic about his own future, but that of the sport, as well. Injured three years ago in a

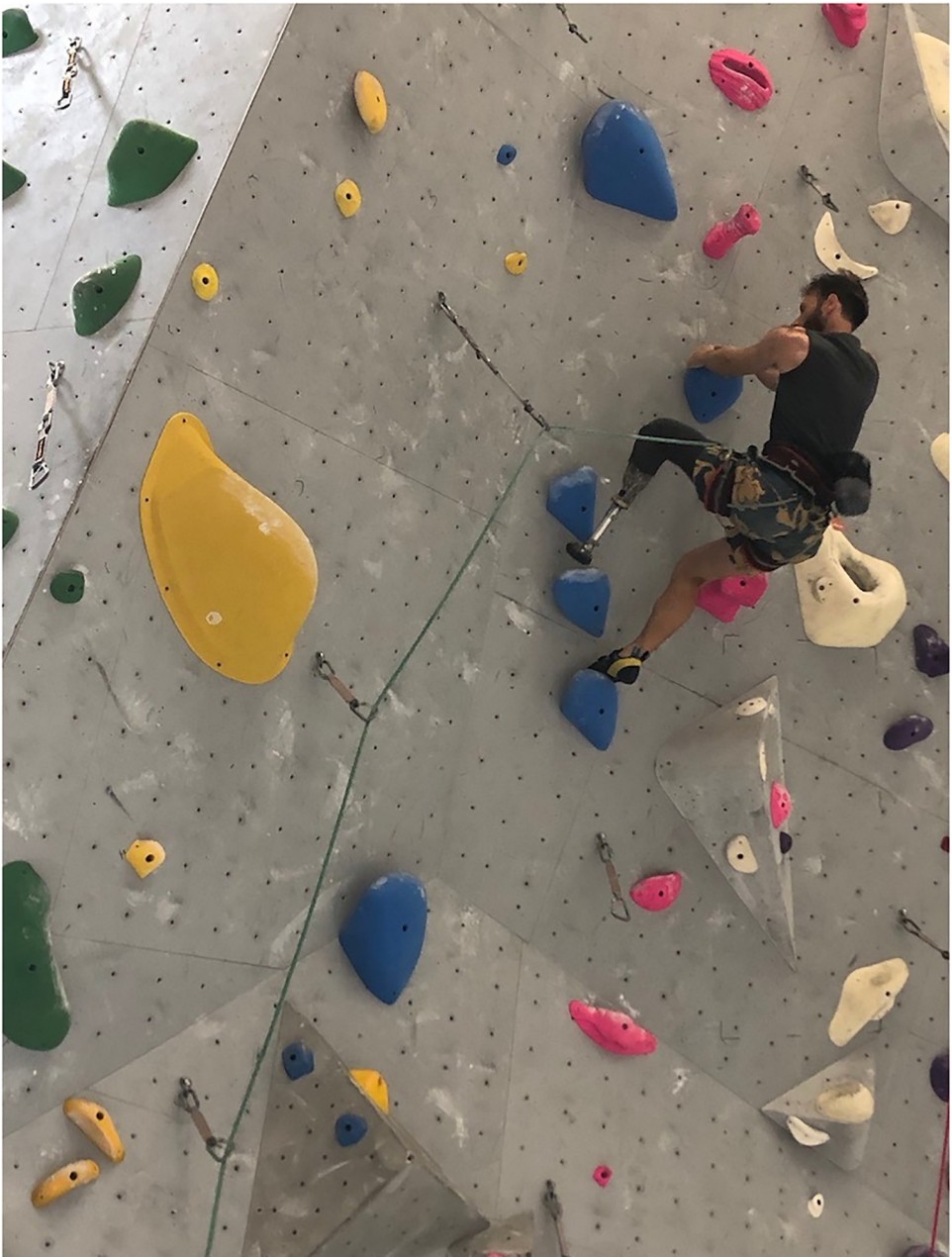

**Fig 1. Jono on a difficult gym route.** He also regularly climbs outdoors.

hiking accident at the age of 19 that resulted in an incomplete spinal cord injury, Tanner now uses a wheelchair and utilizes his upper body muscles to climb in a "relatively unconventional way". As a university undergraduate at the time of this interview, Tanner described himself as being an incredibly competitive person who has always been an athlete. Of his accident, he claims to have "no regrets. The consequences have been positive". Without denying that the risk of injury in rock climbing is higher for those with impairments, the bottom line, he says, is "having fun":

At times, I'm not. . .sure how to feel about it, other than being really excited. It just further validates and further reassures me that. . .the sequence of events that have led up to this point in my life, have been for a reason and have been really incredible. . .but I also see so much more for myself. I feel like I keep saying this, but it's exciting.

Tanner does not always feel accepted and respected by the non-disabled climbing community at the gym in which he works; but he remains optimistic about his chances of competing in the first rock climbing Paralympic competition, which will hopefully be held in 2028:

I'm definitely just very excited to kind of see what's possible. I'm excited to put more effort in and continue my training and just exercise my determination and commitment and all of this just to see what could be.

At a group discussion with the climbing team, I (NS) asked what they would like to see accomplished in the next 25 to 30 years. As a speculative design major, Tanner sees the possibility of universal design leveling the playing field in the future. (see https://www.editorx.com/shaping-design/article/speculative-design).

## Able to push it: "Find[ing] my limits"

*I run for those who think they can't.*

Rick Hoyt, cerebral palsy; Ironman

We continue with two individuals, one who was the victim of a drive-by shooting and has been using a wheelchair for many years, and one newly disabled having had both his kness below the knee amputated due to streptococcal A sepsis.

Cameron and Lingwood [65] note that "when disability is considered as signifying personal loss and limitation. . . it excludes from consideration the possibility that somebody might live with impairment as an everyday part of their life. That they may consider being impaired an important part of who they are, and that it has shaped their life experience and who they have become as a person, and that they might like themselves and enjoy being who they are regardless of their impairment experience." A key takeaway of the affirmative model further supports "the rights of disabled people to be who they are, as they are" (ibid 316).

Eduardo (pseudonym), a 45-year-old wheelchair triathlete, was injured as a bystander in a drive-by shooting in Los Angeles at the age of 17. As a professional sponsored athlete, he travels widely for his sport and sees himself as being "very competitive, competent, and driven". In addition to competing in numerous triathlons throughout the world, Eduardo has completed a 265-mile bicycle race through Alaska. To Eduardo, "living in a wheelchair is no life". When asked whether he has to deal with much pain when competing, he responded that:

Sometimes we have to push the handcycle with one hand. Sometimes you get blisters inside your hands. But you have to keep going. You have to find a way to keep going. Those are the challenges. I was able to push it. You have to always find a way how to do it. . .

In Turner's [2] words, "each individual's life experience contains alternating exposure to structure and. . . transitions." Reliant on his adaptive equipment not just for his sport, but for

his survival, Eduardo, talks about his relationship with his equipment, which he views as an extension of his own body. He notes that:

> I pretty much try to do everything on my own. Don't call me crippled, I'm a guy in a wheel-chair. . .. Once it [the wheelchair], breaks down, then I'm a cripple. . .. Today I had to get up at 5:30. By 8 o'clock I was climbing the hill. I got to the top of the hill and I had a flat tire. I had to change my tire. Those are the challenges of life. But you have to learn how to do that by yourself. . . If I said I couldn't do it, I would still be there. Those are challenges. I pretty much do everything. . .

Turner [2] asserts that "the neophyte in liminality must be a tabula rasa, a blank slate, on which is inscribed the knowledge and wisdom of the group". Here, Eduardo discusses the importance of his role as a mentor to neophytes:

> I'm 45 years old now. How do I see it now? I see it as a person who has life experience in a wheelchair. A person who can help someone who is new in a wheelchair. I see life with more fun. With sport I began to find that I like to be challenged. I'd like to find my limits; to see where I'm at.

> It's a big change for people newly disabled. Three months in rehab they teach you how to take care of yourself. Then after three months you're on your own.

> When I see new people, when I see that I can help them, it pays off for everything that I can do. I keep asking myself "why am I doing this"? Tomorrow I need to get up at 6 AM in the morning to go swim in open water, and cold water. No one's telling me to go do that. I'm telling it to myself I need to get up and go do this.

Tom (pseudonym) is a youthful-looking, middle-aged, white, working-class male who is relatively new to the culture of disability and adaptive athletics. He attended a Challenged Athletes Foundation (CAF) triathlon event for the first time to support a friend who runs an adaptive skiing program. Prior to his illness, Tom engaged in hard manual labor, including driving heavy tractor equipment. Sepsis due to severe illness resulted in the amputation of both his legs above the knee. His trauma and abrupt changes to his social relationships have fostered a profound self-reflection and revaluation of life. Defining himself as Having been a "heavy, heavy, heavy, heavy drinker for a lot of years" in which his social life centered around hanging out in bars, Tom believes an alcoholic lifestyle may have contributed to the development of sepsis. Within the past two years, Tom has undergone 21 surgeries and is still learning to walk using prosthetic legs.

Suddenly forced to experience what it is like to be subjected to 'stereotyping and stigmatization', he nevertheless claims to have a better life post-illness as he makes the transition from the liminal state of being "neither here nor there" [2] to that of taking an active part in the community. Recalling his drinking buddies who would not visit him after his surgeries, Tom has become acutely aware of his transition:

> I was [still] trying to fit in. . . I wanted everybody to love me. I don't want to be that person that's stereotyped or stigmatized. . . I'm not used to this. I've never had to deal with this.

Learning new skills and adapting to a changing body image, he described the transition: "now that I got prosthetics,. . . I don't have a disability; I could walk". At this "betwixt and between" juncture, Tom appears to be in limbo about accepting his dis/ability and the physical

limitations which it brings; assuming, for example, that acquiring prosthetics will return him to the status of being non-disabled. While Tom will never return to his initial non-disabled status, being involved in a supportive adaptive athletics environment is providing an opportunity for him to develop a greater sense of self-confidence, belonging and acceptance of his situation. Tom's wife adds that she is proud of his attitude, saying:

> I mean, he could sit here and lay in bed every day and say 'screw it' and pop a beer and just keep. . . like, not want to do it. But he's been monoskiing and fishing and he bicycles, you know, a ton. "Handcycle", he corrects her.

### Creating *communitas*: "Why I am here"

*Communitas* is greater than simply a sense of community. It is the fulfillment of "a sharing and intimacy that develops among persons who experience liminality as a group," [https://www.dictionary.com/browse/communitas]. The bond that develops amongst adaptive athletes builds upon the intertwining of vulnerability and fortitude. While the focus is kept 'on our strengths', it is the mutual stigmatization and limiting perceptions of dis/ability from which *communitas* grows (Fig 2).

Known for its perfect weather and many beaches, San Diego hosts outdoor adaptive athletic activities throughout the year, including Life Rolls On (LRO), in which up to 500 volunteers and attendees show up to the beach on a given day. Athletic gatherings provide a space for the development of emotional, educational, social and peer support among athletes with dis/abilities. Such gatherings also provide opportunities for volunteers to gain a deeper understanding of the everyday life and the unique attributes and challenges of those they are assisting. Some athletes have known each other, trained and attended events together for years; still others meet for a day and move on with their respective lives, possibly to meet at another event (Fig 3).

As the LRO event was coming to a close, a volunteer waved Ty over to talk with me. "You have to meet this guy", she whispered. It has taken Ty a number of years to get his life and his career back in order after having multiple surgeries and ultimately losing his leg up to the hip from complications due to a motorcycle accident. Now a public speaker, Ty went from "being pushed into the waves" by volunteers, to standing on a moving board on his prosthetic leg, to gaining a spot in the upcoming Paralympic surfing competition. Mesmerized, I spent an hour on the beach listening to him talk about the importance of being both an athlete and a mentor to the adaptive athletes community.

> I started with Life Rolls On in 2017. He [Jesse Billauer; quadriplegic surf champion] started this organization to get people with all levels of ability into the water. Quadriplegic, paraplegic, blind, amputee, cognitive situations. Anything. And I feel like this organization saved my life. And that's why I'm here.

Ty expressed the importance of one's mental attitude in shaping the motivation and ability to create a meaningful life under conditions of impairment. He sums up his experience, thus:

> So that's what I'm working on. It's such. . .a mental thing. It may be a physical disability, but. . .because your physical is limited. . .your mental and emotional. . .takes a hit [as well]. Because. . .like, "I should be able to walk over there. Why is it taking me so long to get a couple yards?" Well, your body is different now. Just do your best not to allow your mind to become disabled as well. You know, just do your best.

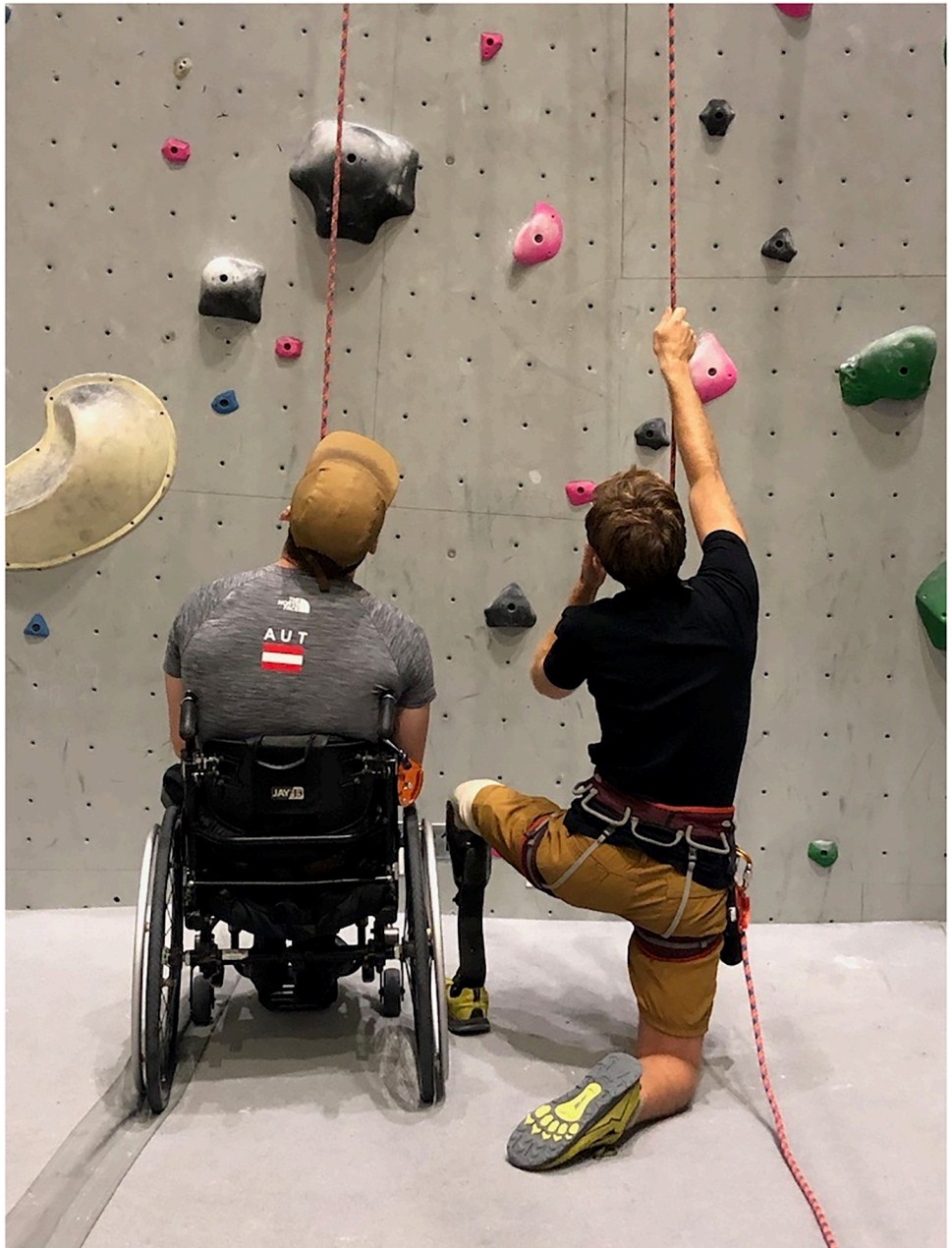

**Fig 2. Communitas.** Tanner and Jono figuring out a route in the gym.

## Discussion: "Focus on our strengths"

"The main interest in life and work is to become someone else

that you were not in the beginning."

Michel Foucault [66]

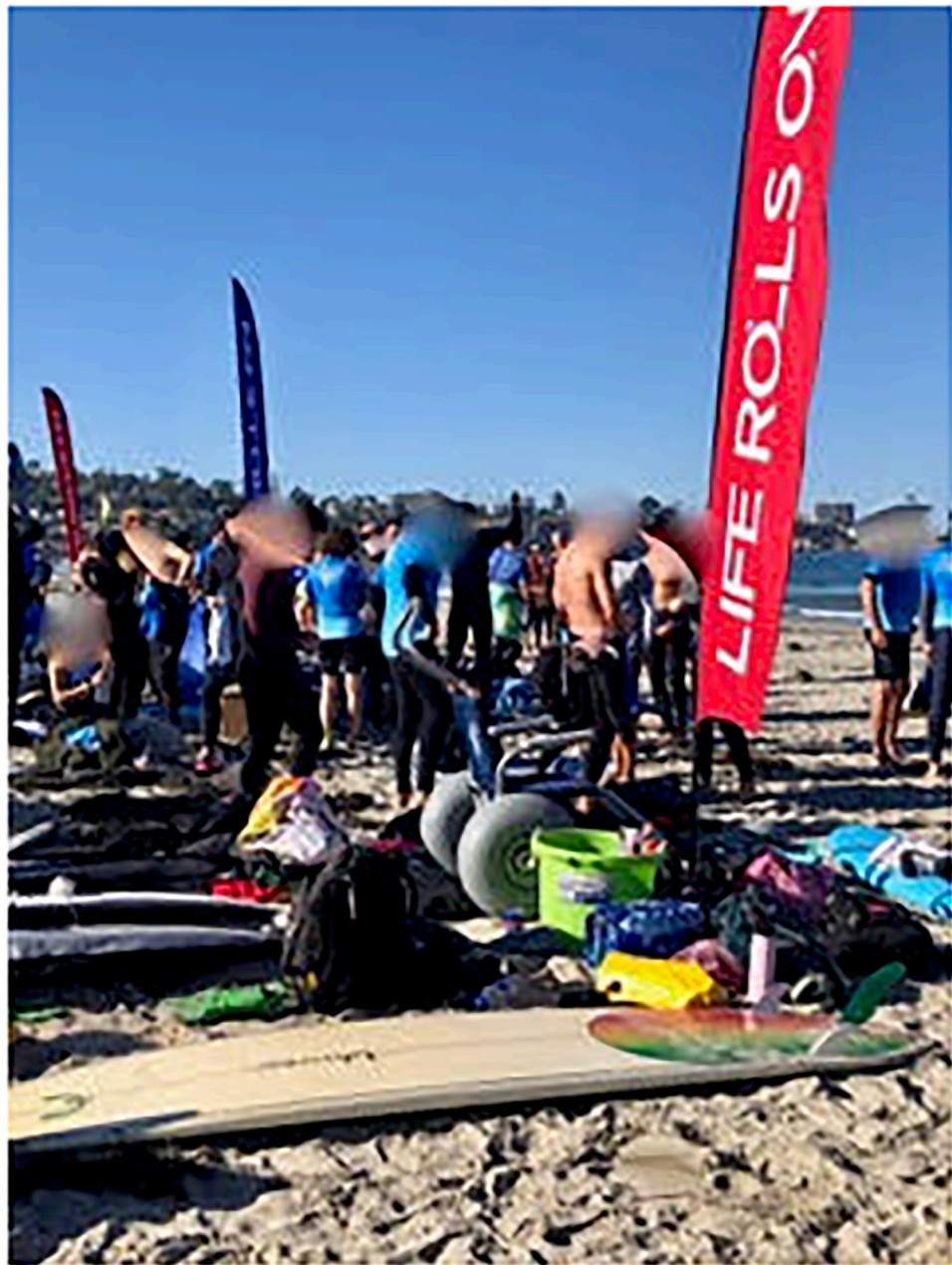

**Fig 3. Volunteers on the beach.** Preparing to help surfers enter the water.

Phenomenologists such as Maurice Merleau-Ponty and Edmund Husserl study the lived experience of the body; specifically, what it is like to be *in* one's body, and *of* one's body [51–54]. *Being* in one's body as an athlete requires, if nothing else, concentration. When rock climbing, unless I am willing to fall, I leave no room in my conscious thought process for anything other than reaching that next hold. When swimming, I count my breaths. I am conscious when in the ocean that I am not in control of the space shared with creatures larger, and probably hungrier, than myself. If I am intent on making it back to shore, it is on my strokes—on my

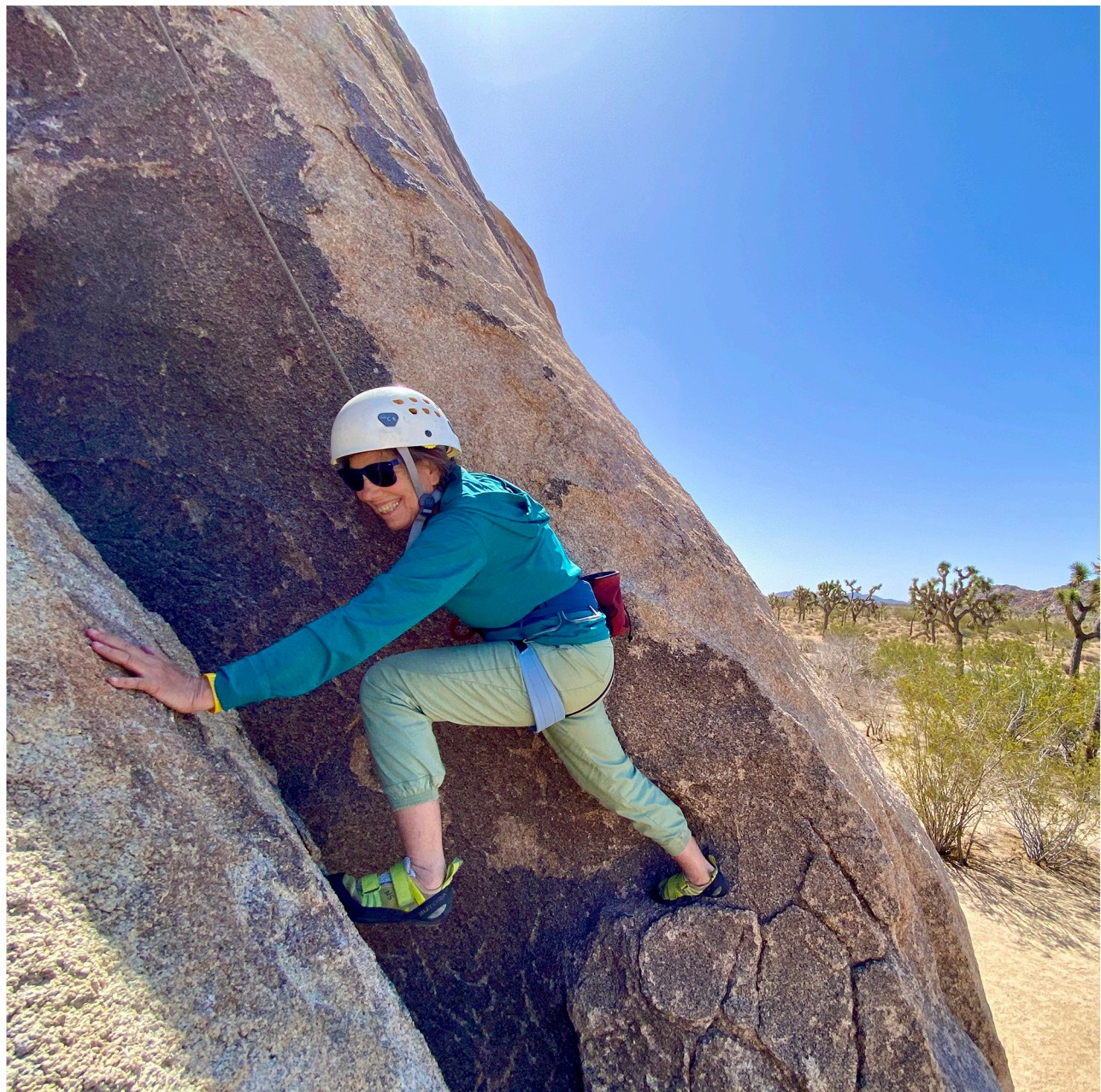

**Fig 4. The author climbing.** Only 3200 more feet to go.

strengths—that I must focus. Even more so if I am to reach my goal of climbing Yosemite's 3200 foot monolith, El Capitan, as one of the few women with a disability (Fig 4).

Through the lens of disability culture, Patrick Devlieger and colleagues [67] introduced the concept of disABILITY Mundus as a worldwide, cross-disciplinary study which encompasses, among other things, such futuristic concepts as the global exploration of 'Transmodern, Trans-human and Posthumanisms' [68]. This rethinking of ability according to Ginsberg and Rapp is key to:

the radical presentation of disability as a resource, and a creative source of culture, that moves disability out of the realm of victimized people, or as an insurmountable barrier [69].

By placing emphasis on *ability*, these authors offer a sense of possibility. We, as humans, have something inside of us that makes us want to excel and be recognized for our accomplishments. For athletes with disabilities, those accomplishments can take various forms. The 'rethinking' of ABILITY as transcending certain physical limitations has been the focus of some of the world's top-achieving adaptive athletes; including a 23 year old man who recently became the first person with Down syndrome to cross the finish line of the onerous Hawaii Ironman. Defying low expectations, Chris Nikic [35] joins the ranks of Rick Hoyt by becoming a renowned author, public speaker and homeowner. Having achieved his goal, he plans to complete the six World Marathon Majors in 2023: New York, Chicago, Boston, London, Berlin, and Tokyo, as well as to advocate for the inclusion of neurodivergent athletes in all the major racing events.

The social emphasis on expanding diversity and inclusion in recreation can not only benefit those with physical impairments, but those without impairments, as well. Says Nico, an administrator with CAF:

Technology has advanced so much. With the prosthetics, a lot of the equipment, you know, the hand cycles, now there are off-road wheelchairs that people can use; so they can access national parks, trails, hiking trails, the beach. So, we really are at a point where things are becoming more universally usable. The idea being that we also have a universal design going forward, right?

Before the advent of the disability rights movement in the 1970's, and the development of motorized and lightweight athletic wheelchairs, prosthetic limbs were hidden under pants, skirts or shirts; rather than being proudly displayed as they are today. With little independence of movement, wheelchair users had to be transferred to chairs and pushed from behind. Wheelchairs and prosthetics today may be artfully decorated and designed to be interchangeable for multiple sports. All-terrain wheelchairs can navigate rocks, sand and even snow. Bionics engineer, Hugh Herr's invention of a bionic leg allows his friend and fellow rock climber, Jim Hewing, to climb with a precision not seen before. Another invention is contributing to a professional dancer who was injured in the Boston Marathon bombing being able to resume her professional career [20, 30].

It is not only professionals who are able to take advantage of these new advances. Ellen (pseudonym) and Craig (pseudonym) showed up to the beach for the surfing event in their new sporty wheelchairs. Having a lightweight all-terrain wheelchair has "changed my life," said Ellen, whose mother used to leave her on the shore and carry buckets of ocean water up to pour over her. Because insurance does not pay for 'luxury wheelchairs', a fund-raising account was opened to help purchase the wheelchair:

By the following spring. . . I had my new bright pink chair and it was absolutely beautiful. A lot of fun. It comes apart and fits in the back of my small van. . .Mama will give me a bit of a push into the wave and it splashes up around me. It's amazing!

"It changed her life", repeats her mother. "It changed her life."

Craigs's sense of autonomy has been transformational, as well, thanks to his all-terrain wheelchair:

It just gives me more independence. . .Before my chair. . .I would have to ask my little brother [for help]. . . I can actually go for a hike now [without] falling out. . .It just offers the stability. . .without worrying about. . .uneven terrain messing you up or messing up the wheelchair. . .This is one of the biggest things for me in terms of independence. I would be able to get out and interact.

During his recovery process, Ty discovered that as he was "just looking for something to get back to normal",

I realized that there really is no getting back to normal; you now are changed forever. . .Trying to fit ourselves into the able-bodied world. . .that's where you can get discouraged. That's where you can, you know, you can get into some really dicey places.

Nico, who directs a veterans training program at CAF, discusses the life changing experiences of the wounded warriors with whom he works:

Oftentimes when I meet these vets, . . .they're at the height of their health when they're in the military, and in a millisecond, they lose their limbs, they're blinded, they're paralyzed, thinking, "What's next? What is my purpose now? What is my value?" And we let them know they're gonna do all the things they used to do and things they never dreamed of doing. . .They know they have an entire team behind them. . .And that gives them a sense of purpose again. . .There's no such thing as a life-limiting ability. . .We want people to get out of their comfort zone.

Attending adaptive athletic events can be fun, fulfilling and educational; even challenging. But, it is not just for the surfing or the climbing or the beach parties that we show up. And, it is not just for ourselves. It is for those who will come after us—those 'not yet dis/abled' individuals, as Judy Heumann might describe them—who will someday rely on our strengths and experience to help them through their respective traumas, that we must continue showing up. It is for those stuck in the 'liminal spaces', as Joan Ablon phrased it—those who might still be struggling with accepting their dis/abilities, that we *keep* showing up. That is where our collective strengths truly lie.

## Conclusion

This exploratory ethnography addressed some of the common misconceptions and stereotypes about athletes with dis/abilities. In it, we applied Swain and French's (2000) affirmative model, which posits a non-tragic view of disability, as well as Turner's [2] theories on liminality and *communitas*. We present some of the voices of a new generation of people living with impairment, born into a world where others who came before fought for the early elements of what has come to be known as universal design, and who fought against stereotypes and limits imposed by society and its institutions. This generation is not deterred from living a fully realized life that can include recognition both at the table and on the podium.

## Acknowledgments

We thank the study participants for sharing their thoughtful responses, openness, and candor in describing their experiences as adaptive athletes. We also thank the organizations which have so generously contributed to the support of adaptive athletic endeavors: The Challenged Athletes Foundation; Surfing Madonna and Life Rolls On. NS would also like to

wholeheartedly thank the participants for accepting her as one of their own, and to CAF for continuing to pay for her rock climbing obsession.

We are especially grateful to Professors Carole Browner and Joan Ablon for their insightful comments on earlier drafts of this article. We furthermore acknowledge Joan Ablon's foundational leadership in the field of disability scholarship, and NS particularly thanks her, not just for her patience in reading innumerable drafts of papers, but also for being an extraordinary mentor, a friend, and a confidant. NS also thanks her co-author (CvG) for her willingness to listen to endless stories and learn about adaptive athletes.

## Author Contributions

**Conceptualization:** Norah Anita Schwartz.

**Data curation:** Norah Anita Schwartz.

**Formal analysis:** Norah Anita Schwartz, Christine Alysse von Glascoe.

**Investigation:** Norah Anita Schwartz.

**Methodology:** Norah Anita Schwartz, Christine Alysse von Glascoe.

**Project administration:** Norah Anita Schwartz.

**Supervision:** Norah Anita Schwartz.

**Validation:** Norah Anita Schwartz, Christine Alysse von Glascoe.

**Writing – original draft:** Norah Anita Schwartz, Christine Alysse von Glascoe.

**Writing – review & editing:** Norah Anita Schwartz, Christine Alysse von Glascoe.

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
