## [Decision Letter · Decision Letter 0]

3 Jan 2023

PONE-D-22-29815Adaptive sport as affirmation: “We focus on our strengths, not our disabilities”PLOS ONE

Dear Dr. Schwartz,

Thank you for submitting your manuscript to PLOS ONE. After careful consideration, we feel that it has merit but does not fully meet PLOS ONE’s publication criteria as it currently stands. Therefore, we invite you to submit a revised version of the manuscript that addresses the points raised during the review process.

We look forward to receiving your revised manuscript.

Kind regards,

Yih-Kuen Jan, PhD

Academic Editor

PLOS ONE

Journal Requirements:

4. We note that Figures 1, 2, 3, 4, 5 and 6 in your submission contain copyrighted images. All PLOS content is published under the Creative Commons Attribution License (CC BY 4.0), which means that the manuscript, images, and Supporting Information files will be freely available online, and any third party is permitted to access, download, copy, distribute, and use these materials in any way, even commercially, with proper attribution. For more information, see our copyright guidelines: http://journals.plos.org/plosone/s/licenses-and-copyright.

a. You may seek permission from the original copyright holder of Figures 1, 2, 3, 4, 5 and 6 to publish the content specifically under the CC BY 4.0 license. 

Reviewers' comments:

Reviewer's Responses to Questions

**Comments to the Author**

1. Is the manuscript technically sound, and do the data support the conclusions?

Reviewer #1: Yes

Reviewer #2: Partly

2. Has the statistical analysis been performed appropriately and rigorously? 

Reviewer #1: N/A

Reviewer #2: No

3. Have the authors made all data underlying the findings in their manuscript fully available?

Reviewer #1: Yes

Reviewer #2: Yes

4. Is the manuscript presented in an intelligible fashion and written in standard English?

Reviewer #1: Yes

Reviewer #2: Yes

5. Review Comments to the Author

Reviewer #1: This manuscript is well written and presents a perspective about athletes with disabilities that is needed and rarely described. In a scientific field that often addresses pathology, this paper addresses the internal motivations and self-assessments of athletes who reject the limitations of the labels.

Reviewer #2: The reviewer applaud researchers for researching on this rare area. However, Introduction should be short. Reviewer could not pick the difference between introduction and background to study. The background to study should show the problem and its setting. researchers should revisit it and unpack a bit. Also, research questions can help in focusing during data collection. With research questions, findings can be more focussed.

Research findings are detailed and real as they were sourced verbatim from horse's mouth. However, whilst results are good, they tend to be too detailed on each participant. Most sound like biodata. More so, there is a tendancy of over emphasising on certain participants, for example,Jesse Billauer Edwardo to mention a few. Bring out from them key issues relevant to your research questions. More so if these are not pseudor names, there is need to seek consent of publishing their names and their nature of disability so that their confidentiality is not violated.

On another note, the discussion is an extention of findings. there is need to revisit it. Discussion should marry literature and findings such that conclusion is 'born' out of it. Thus there is need to revisit this area too.

On a general point of view, the research is so good. What is needed is to visit areas pointed out and amend.

6. PLOS authors have the option to publish the peer review history of their article (what does this mean?). If published, this will include your full peer review and any attached files.

Reviewer #1: No

Reviewer #2: **Yes: **Dr Gilliet Chigunwe

---

## [Author Response · Author response to Decision Letter 0]

23 Feb 2023

Yih-Kuen Jan, PhD

Academic Editor

PLOS ONE

February 10, 2022

Dear Dr. Yih-Kuen Jan: 

This letter describes the changes we made in response to the points raised during the review process of our submission, PONE-D22-29815, Adaptive sport as affirmation: “We focus on our strengths, not our disabilities”. 

Reviewer #1: “This manuscript is well written and presents a perspective about athletes with disabilities that is needed and rarely described. In a scientific field that often addresses pathology, this paper addresses the internal motivations and self-assessments of athletes who reject the limitations of the labels.”

Changes:

 This reviewer did not require any changes. 

Reviewer #2, Comment 1: “The reviewer applaud researchers for researching on this rare area. However, (a) Introduction should be short. Reviewer could not pick the difference between introduction and background to study. (b) The background to study should show the problem and its setting. researchers should revisit it and unpack a bit. Also, (c) research questions can help in focusing during data collection. With research questions, findings can be more focussed.”

Changes:

a. We substantially shortened the introduction, making a clearer separation from the background of the study. P3.

b. We made significant changes to the background, addressing issues brought to our attention by readers of an earlier draft. Thus, we included more background on Victor Turner’s theories of liminality and communitas.Because people with impairments often feel like outsiders and newly disabled people have their lives suddenly turned around by trauma, we felt that these theories would be the most fitting. As one moves from the sense of being alone and excluded to that of being accepted by a community, they enter the new phase of communitas. P2

c. We added research questions. P3. Specifically, we added the following paragraph: 

As part of a larger ethnographic project, the research questions aim to address the role of athletics in the lives of adaptive athletes. Specifically, we aim to understand the unique challenges, benefits and personal growth experiences that participants face in their chosen sport(s). What challenges, for example, does a newly disabled person face when entering or re-entering athletics? How does participating in team sport alter one’s self-perception and view of life? For parents of children with impairments, how does being a part of an adaptive athletic community help their children grow and thrive? What is the range of emotions experienced by athletes and their families?

Reviewer 2, Comment 2: “Research findings are detailed and real as they were sourced verbatim from horse's mouth. However, whilst results are good, they tend to be too detailed on each participant. Most sound like biodata. More so, there is a tendency of over emphasising on certain participants, for example, Jesse Billauer Edwardo to mention a few. Bring out from them key issues relevant to your research questions. More so if these are not pseudor names, there is need to seek consent of publishing their names and their nature of disability so that their confidentiality is not violated.”

• We shortened the details on each participant without losing the context of their lived experiences. We spread out the emphasis to equalize the focus on each of the participants. Pp 15-26.

• We received consent to publish first names and photographs of a select group of participants. We used pseudonyms without photographs for the others. 

Reviewer 2, Comment 3: “On another note, the discussion is an extension (sic) of findings. there is need to revisit it. Discussion should marry literature and findings such that conclusion is 'born' out of it. Thus there is need to revisit this area too.”

We rewrote the discussion section in order to bring it into better alignment with the literature and the findings, and to lead to the conclusion. In revisiting the discussion, we decided to include a more theoretical framework within the paper and discuss the work on liminality and communitas as theorized by anthropologist Victor Turner and fitting our ethnographic findings on adaptive athletics. 

We have reviewed the reference list to ensure that it is complete and correct. We have not added or deleted any retracted references. 

We appreciate your kind review of our manuscript and look forward to addressing any further issues that might need attention.

Sincerely,

 Norah Anita Schwartz, Ph.D., MPA

Christine Alysse von Glascoe, Ph.D., M.D.

---

## [Decision Letter · Decision Letter 1]

20 Mar 2023

Adaptive sport as affirmation: “We focus on our strengths, not our disabilities”

PONE-D-22-29815R1

Dear Dr. Schwartz,

We’re pleased to inform you that your manuscript has been judged scientifically suitable for publication and will be formally accepted for publication once it meets all outstanding technical requirements.

Kind regards,

Yih-Kuen Jan, PhD, University of Illinois at Urbana-Champaign

Additional Editor Comments (optional):

Reviewers' comments:

Reviewer's Responses to Questions

**Comments to the Author**

1. If the authors have adequately addressed your comments raised in a previous round of review and you feel that this manuscript is now acceptable for publication, you may indicate that here to bypass the “Comments to the Author” section, enter your conflict of interest statement in the “Confidential to Editor” section, and submit your "Accept" recommendation.

Reviewer #1: All comments have been addressed

Reviewer #2: All comments have been addressed

2. Is the manuscript technically sound, and do the data support the conclusions?

Reviewer #1: Yes

Reviewer #2: Yes

3. Has the statistical analysis been performed appropriately and rigorously? 

Reviewer #1: N/A

Reviewer #2: N/A

4. Have the authors made all data underlying the findings in their manuscript fully available?

Reviewer #1: Yes

Reviewer #2: Yes

5. Is the manuscript presented in an intelligible fashion and written in standard English?

Reviewer #1: Yes

Reviewer #2: Yes

6. Review Comments to the Author

Reviewer #1: This manuscript addresses and important set of considerations for the understanding of disability. I recommend that it be published.

Reviewer #2: Researchers have addressed the issue of consent & confidentiality. Good. Such a stance protects participants especially those whose disabilities they don't want to be published.

7. PLOS authors have the option to publish the peer review history of their article (what does this mean?). If published, this will include your full peer review and any attached files.

Reviewer #1: No

Reviewer #2: No

---

## [Editor Report · Acceptance letter]

19 Apr 2023

PONE-D-22-29815R1 

Adaptive sport as affirmation: “We focus on our strengths, not our disabilities” 

Dear Dr. Schwartz:

I'm pleased to inform you that your manuscript has been deemed suitable for publication in PLOS ONE. Congratulations! Your manuscript is now with our production department. 

Kind regards, 

on behalf of

Dr. Yih-Kuen Jan 

Academic Editor

PLOS ONE